# An Appropriate Genetic Approach for Improving Reproductive Traits in Crossbred Thai–Holstein Cattle under Heat Stress Conditions

**DOI:** 10.3390/vetsci9040163

**Published:** 2022-03-28

**Authors:** Akhmad Fathoni, Wuttigrai Boonkum, Vibuntita Chankitisakul, Monchai Duangjinda

**Affiliations:** 1Department of Animal Science, Faculty of Agriculture, Khon Kaen University, Khon Kaen 40002, Thailand; akhmad.fathoni@ugm.ac.id (A.F.); wuttbo@kku.ac.th (W.B.); vibuch@kku.ac.th (V.C.); 2Department of Animal Breeding and Reproduction, Faculty of Animal Science, Universitas Gadjah Mada, Yogyakarta 55281, Indonesia; 3Network Center for Animal Breeding and OMICS Research, Khon Kaen University, Khon Kaen 40002, Thailand

**Keywords:** fertility, genomic selection, heat tolerance, tropical Holstein

## Abstract

Thailand is a tropical country affected by global climate change and has high temperatures and humidity that cause heat stress in livestock. A temperature–humidity index (THI) is required to assess and evaluate heat stress levels in livestock. One of the livestock types in Thailand experiencing heat stress due to extreme climate change is crossbred dairy cattle. Genetic evaluations of heat tolerance in dairy cattle have been carried out for reproductive traits. Heritability values for reproductive traits are generally low (<0.10) because environmental factors heavily influence them. Consequently, genetic improvement for these traits would be slow compared to production traits. Positive and negative genetic correlations were found between reproductive traits and reproductive traits and yield traits. Several selection methods for reproductive traits have been introduced, i.e., the traditional method, marker-assisted selection (MAS), and genomic selection (GS). GS is the most promising technique and provides accurate results with a high genetic gain. Single-step genomic BLUP (ssGBLUP) has higher accuracy than the multi-step equivalent for fertility traits or low-heritability traits.

## 1. Introduction

As global temperatures have risen, heat stress in dairy cattle has become a severe concern for the worldwide dairy sector, especially in tropical countries. The average temperature on Earth increased by 1.09 °C during 2011–2020 [1]. Thailand is one of the countries that has experienced climate change. Thailand has a tropical climate with high relative humidity and high ambient temperatures. Thailand’s average temperature is 35.0–39.9 °C, with 72–74 percent relative humidity [2]. According to the Thai Meteorological Department, the annual mean temperature in Thailand increased by about 1.50 °C from 1970 to 2007 [3]. Hence, livestock in Thailand faces heat stress, especially dairy cattle.

Heat stress is caused by changes in the environment around the animal, including temperature, relative humidity, solar radiation, and wind speed [4]. The high temperature and humidity in Thailand may be the main factors causing heat stress in livestock. Livestock cannot dissipate body heat because of higher environmental heat intensity [5]. This causes animals to pass through the range of the thermoneutral zone, and heat stress will occur [6]. Heat stress also affects livestock reproduction. The direct effects are changes in endocrine regulation, interference of ovarian activity concerning steroid hormone synthesis, and altered progesterone concentrations [2]. Consequently, heat stress reduces estrous activity and impairs fertility performance, including long days open, a greater length of the calving interval, and a low conception rate [7]. Heat stress also has various adverse effects on milk yield and milk quality, thus harming farmers financially [8]. Optimizing feed and housing conditions are two techniques for minimizing the consequences of heat stress [9]. Although these methods can be successful in the short term, they may not be adequate for long-term strategies. Improving genetics, identifying and breeding animals with a natural resilience to extreme heat and humidity and establishing management approaches for long-term solutions should be performed in concert.

Understanding the genetic response to heat stress could help improve the management of feed or housing modifications for dairy cows. Ravagnolo and Misztal [10] proposed a strategy for examining the impact of heat stress on the genetic response of dairy cows using weather station data to predict the level of heat stress tolerance on the performance of dairy cattle in the United States. Tolerance to heat stress is the adaptive process that allows animals to endure the impacts of rising ambient temperature beyond the thermal neutral zone’s temperature–humidity index (THI) limit [11]. The THI limit value at which performance begins to decrease can determine the level of heat tolerance in animals. A THI value of 72 for production, and roughly 68 for reproduction was considered the threshold for heat stress [12].

Genetic differences in thermal tolerance between animals provide clues for selecting animals that are more tolerant to heat stress [13]. Previous studies on heat tolerance found genetic variation in the reproductive performance of dairy cattle under heat stress conditions [9,14,15]. Therefore, genetic improvement via the identification and selection for tolerance to heat in animals for reproductive traits is possible. In general, the genetic improvement of dairy cattle is cost-effective since it results in long-term and cumulative change [16]. However, due to low heritability values and large generation intervals, the traditional method’s rate of genetic improvement is low [17]. Compared to traditional breeding, genome selection is more appropriate for selecting heat tolerance, as it allows faster genetic gain [11]. To predict the breeding value of selection animals, genomic selection uses single-nucleotide polymorphisms (SNPs), genetic markers found throughout the genome [18]. This study aims to investigate the appropriate genetic approach for improving reproductive traits in Thai–Holstein crossbreeds under heat stress conditions.

## 2. Thai–Holstein Crossbred Cattle

In Thailand, dairy cattle are primarily crossbreeds of *Bos taurus* (Holstein, Jersey, Brown Swiss, and Red Dene) and *Bos indicus* (Sahiwal, Red Sindhi, Brahman, and Thai Native). However, most (>95%) are crosses between Holstein and Sahiwal cattle or Thai Native breeds [9]. According to the Thai Department of Livestock Development, Thailand’s dairy cattle population was reported to be 818,537 heads in 2021, with 358,500 cows on 24,764 farms, producing roughly 3000 tons of raw milk each day [19,20]. Most dairy farmers in Thailand (>80%) are smallholders, with only a few large commercial dairy farms [19].

The dairy cattle breeding improvement program in Thailand has been carried out for 60 years. In the beginning, both live purebred dairy cows and frozen semen were imported from subtropical countries, followed by an artificial insemination (AI) program in 1956. Thai Native cattle were inseminated with that semen, and now, they are being developed as crossbred Thai dairy cattle, with Holstein Friesian (HF) selected as the predominant breed [20]. The purpose of crossing those two breeds was to improve milk production, reproduction, and the tolerance of heat, tick-borne diseases, and other diseases of zebu cattle [21,22]. The main population in Thailand is now dairy crossbreds with ≥87.5% HF blood levels [20].

Currently, the development of crossbred Thai–Holstein cattle is still being carried out to select cattle with good performance and suitable for tropical conditions. Genetic evaluation of their economic traits is performed through a progeny testing program by the Thai Department of Livestock Development (DLD) [20].

## 3. Reproductive Traits in Dairy Cattle

### 3.1. Interval Traits

#### 3.1.1. Age at First Calving (AFC)

AFC is the age at which heifers first calve and is also known as the start of the heifer’s productive life [23,24]. AFC influences the lifetime production, reproduction of the female, milk production, and lifetime calf crop [25]. Indirectly, AFC affects the cost invested for rearing cattle. The recommended age for the first calving in local breeds is three years, with many experts recommending between 23 and 25 months, which is regarded as optimal for increasing the dairy company’s profitability [24]. Previous studies on Thai–Holstein crossbred cattle reported an average AFC value of 32.39 ± 5.76 and 30.97 ± 6.65 months [26,27], higher than the recommended age. The same results were reported in Friesian × Zebu (F1), and Friesian × Arsi (F1) cattle were obtained as 35.3 and 33.9 months, respectively [25]. The lower values of AFC reported in US dairy cattle were 24.5 ± 2.73 months for Holstein, 22.9 ± 2.74 months for Jersey, and 26.3 ± 3.13 months for Brown Swiss [28]. The reports indicate that crossbreed heifers reach puberty later than pure breed cows. The variation in the reproductive performances of dairy cattle such as AFC is affected not only by their genetics but also by other aspects, including diet, management, health, and the environment. In Thailand, the AFC is not used as a selection criterion because it is entirely influenced by farmer management to determine the age and body condition score (BCS) at first service. However, AFC and body weight (BW) at calving have been found, in several studies, to have a considerable impact on future milk production and herd survival [29,30,31]. A greater BW or body condition score (BCS) at calving improves milk production later [29]. In contrast, excessively increased fat deposition throughout the raising period, resulting in heavier heifers at first calving, may compromise mammary development, reducing future milk production [32]. The long-term impact of AFC on reproductive success is still unclear. Calving heifers between the ages of 25 and 26 months had shorter recurrent calving intervals than heifers between the ages of 24 and older age groups [30].

#### 3.1.2. Calving to First Service Interval (CFI)

The number of days between calving and the first artificial insemination or natural service in cows is defined as the CFI [33]. The recommended value of CFI in dairy cattle for good management is less than 70 days [34]. The CFI in Thai–Holstein crossbred cattle reached 101.06 ± 44.88 days [26], higher than Norway dairy cattle of 62.5 days and Japanese dairy cattle of 84.9 days [33,35]. In Arsi Negele, Ethiopia, for crossbred dairy cows, the value of CFI was reached at 184 days [36]. Those variations might be due to different genetic architectures, the environment, herd management and practices, and disease. Similarly, Softic et al. [37] also reported that extended CFI was influenced by several factors such as nutrition, endometritis, and a lack of estrus detection. The other factor, uterine infection, has been associated with extended CFI in dairy cattle and low reproductive performance [33,38]. Poor management techniques, such as failing to detect the presence of estrus in dairy cows, can also contribute to a long CFI [25].

#### 3.1.3. Interval from First to Last Insemination (IFL)

The IFL is defined as interval days between the first and last service. It is one of the indicators used by farmers to cull cows after a series of inseminations because it can increase production costs [39]. The IFL days tend to be prolonged with the increasing number of services per conception (NSPC) in cattle. The IFL in Thai–Holstein crossbred cattle was reported to be 49.70 days [26]. The value is nearly the same as the IFL reported in Czech Holsteins of 43.95–50.13 days [40]. A lower IFL value was found in Japanese Black cattle, 28 days.

#### 3.1.4. Days Open (DO)

DO is the time period between calving and successful conception. It is also the part of the calving interval (CI) that can be shortened by improved dairy management and genetics [25]. As a result, DO is one of the indicator variables most widely used to assess dairy cattle fertility. A long DO will cause a prolonged CI and can affect the overall economic income of farmers. Previous researchers reported that the average DO in Thai–Holstein crossbred cattle was 149.93 ± 78.60 days, 152 days, and 156.44 ± 20.77 days [9,26,41]. The other Frisian crossbreeds also have a higher DO, such as Friesian × Arsi (189 ± 9 days), but this did not occur in Friesian × Horro cattle (91.5 ± 1.20 days) and Jersey × Horro cattle (79.20 ± 3 days), which have lower values of DO [42,43]. Pure Holstein cattle are also reported to have a moderate DO of 139.5 days [4]. The high number of DO will affect the extra costs incurred per day [44]. On US dairy farms, an increase in DO from 112 to 166 causes economic losses of 3.2 USD to 5.1 USD per cow per day [45]. Dairy farmers of 44 farms in Central Thailand have to spend as much as 268,088 baht (8220.37 USD) for improving the reproductive performance program every production period [46].

#### 3.1.5. Calving Interval (CI)

The CI is divided into three periods: postpartum anestrus, service period, and gestation [24]. The ideal CI is 12 to 13 months, assuming an average gestation period of 280 days and 85 days for DO [23]. The CI is one of the major aspects of reproductive performance that impacts the livestock production system [47]. The CI in Thai–Holstein crossbred cattle has been reported by a previous researcher to be 14.27 ± 2.62 months [26] higher than Friesian Holsteins of 13.12 months, Friesian × Arsi of 14.23 months, Ayrshire of 13.32 months, Guernsey of 13.10 months, and Jersey of 12.92 months [48,49]. An extended CI can result in a reduced milk yield, reduced lifetime productivity, and increased culling rates and replacement cost [23,35]. A prolonged CI may mainly be caused by a longer DO due to environmental variables such as management practices (poor nutrition, poor housing, estrus detection, semen handling), postpartum problems, and genetics [3,22]. Reproductive health problems result in uterine infection, delayed postpartum estrus, decreased fertility, and an extended CI [43].

### 3.2. Binary Traits

#### 3.2.1. First Service Conception Rate (FSCR)

The FSCR is the percentage of cows that become pregnant after calving at the first insemination (35). The values for the FSCR in Thai–Holstein crossbred cattle were reported to be 54% and 46% [26,50], similar to dairy crossbred cattle in Ethiopia of 41 to 45% [51]. Haile and Yoseph [35] reported that the FSCR was significantly affected by the parity number and season of calving. Similar results were reported by Hammoud et al. [52] and Lemma et al. [53], who associated enhanced fertility with an appropriate ambient temperature and the availability of forage.

#### 3.2.2. Conception Rate (CR)

Most Interbull-participating countries divide the CR into three categories: the heifer conception rate (HCR), cow conception rate (CCR), and daughter pregnancy rate (DPR). The HCR is defined as the percentage of inseminated heifers that become pregnant at each service [54]. The CCR is a measure of the female’s fertility, whereas the SCR is a measure of the bull’s fertility [55]. The optimum recommended range of CR in dairy cattle is 80–85% [35]. CCR in dairy cattle in Thailand is reported to be low, at <40% [56]. In contrast, the CCR value of crossbred dairy cows in Ethiopia’s central highlands was previously found to be 79.3% in a previous study [57]. Haile and Yoseph [35] reported that the CR was significantly (*p* < 0.05) affected by the season of calving and parity. The CR of cows in the rainy season was higher than in dry rainy seasons; this might be because of moderate climatic conditions and the availability of forage.

#### 3.2.3. Pregnancy within 90 Days after First Service (P90)

The successful insemination date within 90 days of the first insemination is referred to as P90. Previous studies showed that the values of the P90 in Thai Crossbred cattle were 75% and 82.17% [26,50], lower than Spanish dairy cattle of 83% [58]. The previous report by Buaban et al. [50] and González-Recio and Alenda [58] showed that the P90 had positive genetic correlations with FSCR and test-day milk yield ranging from 0.29 to 0.69 and 0.92 to 0.97. Otherwise, a negative genetic correlation was found in the P90 inseminations per lactation (INS) of −0.54 [58].

### 3.3. Count Traits

#### The Number of Services per Conception (NSPC)

The number of services required for a successful conception is referred to as the NSPC [57]. The NSPC is a measure of reproductive efficiency that expresses the fertility of dairy cows [59]. In a dairy farm, the NSP largely depends on the breeding program used. A lack of knowledge and experience in the timing of insemination, unqualified technicians, and disease and hygiene problems are the most common reasons for a high rate of the NSCP [60]. In natural mating, the NSPC is less controlled than using artificial insemination (AI), and NSPC values higher than 2 can be considered poor [43]. A previous study reported that natural service conception has lower values (1.18) than artificial insemination users (1.5 up to 2.3). This might indicate that many environmental factors affect the AI method, including the availability of feed, the insemination time, and estrus detection in cattle [24]. The average values of the NSCP of Thai–Holstein crossbred cattle were reported to be 2.29 ± 1.72 and 1.90 ± 1.37 [26,50]. Those were higher than dairy cows in midland and low-land areas in Ethiopia reported as 1.54 ± 0.55 and 1.82 ± 0.65, respectively [60]. Different locations may offer different management and affect the NSPC. Feed and high environmental temperature were also reported to have an effect on the NSPC [59]. NSCP also has an impact on farmers’ economies. In Korea, dairy calves that were unable to conceive cost an additional USD 55.40 for reproductive treatment and palpation [61].

## 4. Heat Stress and Its Effect on Reproductive Traits

Heat stress is one of the significant factors affecting fertility [62]. Heat stress is a physiological condition in which an animal’s body temperature rises over its typical range of activity due to total heat gain from both internal and external sources [63]. Hence, evaporative cooling through sweating is a significant part of the heat-dissipation process in a high-temperature environment [64]. Since sweating plays an important role in the mechanism of heat dissipation, skin morphology must also be involved. Previous studies have found effects between different breeds of Sahiwal (*B. indicus*), HF (*B. taurus*), crossbred with 75% and 87.5% HF, respectively, and the interaction effect between skin color and the genetic fraction of HF on skin morphology [65]. The study revealed that Sahiwal had the highest density and volume of sweat glands, as well as the highest density of hair follicles (1058 glands/cm^2^; 1.60 μ^3^ × 10^−6^) compared with pure HF (920 glands/cm^2^; 0.51 μ^3^ × 10^−6^ and crossbred with 75% and 87.5% HF (709 glands/cm^2^; 0.68 μ^3^ × 10^−6^; and 691 glands/cm^2^; 0.61 μ^3^ ×10^−6^), respectively (*p* < 0.01). However, the components of skin blood flow, such as the capillary diameter, capillary circumference, and capillary surface, were higher in the HF purebreds (8.33 μm, 26.48 μm, and 2.07 μm per cm^2^, respectively), 87.5% HF (7.13 μm, 22.40 μm, and 1.95 μm per cm^2^, respectively), and 75% HF (7.85 μm, 24.92 μm, 1.83 μm per cm^2^, respectively) than in the Sahiwal (7.24 μm, 22.49 μm, 1.79 μm per cm^2^) (*p* < 0.01). This suggests that skin morphology may influence cutaneous evaporative heat-loss capabilities and heat tolerance in crossbred cattle.

The level of heat tolerance in animals can be determined by the THI limit, where livestock performance begins to decline. The THI value is used to assess heat stress levels in dairy cattle by combining the effects of temperature and humidity [2]. The THI chart for estimating heat stress levels in dairy cattle is presented in Figure 1. The THI in Thailand could reach the highest range (82.62–82.80) from March to May [2]. Therefore, this situation can cause heat stress and has an impact on livestock performance, especially in reproductive performance.

The effects of heat stress on livestock reproduction altered endocrine regulation in the hypothalamic–pituitary–ovarian axis (HPO axis), impaired ovarian activity, and impaired progesterone concentrations, which reduced estrous behavior and caused a prolonged number of days open and low conception rate [7,67]. Kornmatitsuk et al. [68] reported that the days open for Thai dairy cattle in the hot season were higher (111.2 ± 8.9) than in the cool season (97.4 ± 5.8). The reduction in CR during heat stress can be up to 30%, with a clear seasonal pattern of estrus detection [62]. Increased ambient temperatures have a detrimental impact on the cow’s ability to perform natural mating behavior by reducing the duration and intensity of estrus expression [69].

A previous study stated that heat stress also affected the service period, FSCR, CFI, and CI in dairy cattle. The cows that calved under heat stress conditions had a longer service period, 299 ± 11 days, than in the normal condition, reported as 133 ± 7 days [70]. The FSCR rate during the cool season in a commercial farm of Thailand was also reported as higher, 23.1%, than the hot season, 9.8% [68]. In the same way, Kananub et al. [71] stated that the FSCR in the winter season had a three-times greater chance of success compared to the summer season. In Argentina, the CFI in the cows that received their service during the hot season was longer (97 days) than those that had their service from October to December (83 days) [72]. In North Africa, cows calving in the absence of heat stress (THI ≤ 70) have a shorter CI of 420 ± 15.1 days than cows calving in the presence of heat stress (THI ≥ 80) of 487 ± 12.8 days [73].

## 5. Genetic Parameters and a Model for Reproductive Traits in Dairy Cattle Experiencing Heat Stress

### 5.1. Heritability (h^2^)

Heritability is the difference in the phenotype between animals caused by genetic factors. Broad-sense heritability is defined as the proportion of phenotypic variation (VP) due to genetic values (VG), which may include dominance (VD) and epistasis (VI) effects. Narrow-sense heritability, however, only represents the proportion of genetic variation owing to additive genetic values (VA) [74]. Narrow-sense heritability is commonly utilized in animal breeding because the response to artificial (and natural) selection is dependent on additive genetic variation. Furthermore, additive genetic variation is the primary driver of relative similarity [74].

Heritability ranges from 0 to 1. A higher value of heritability means that most of the differences between the phenotypes of animals are genetic [75]. In contrast, while a low heritability indicates a low VA, it says a lot about VG, but it does not account for total genetic variance. As a result, a character h^2^ might still have a lot of genetic diversity at the loci that contribute to the observed trait variance. When a variable has a heritability value of near zero (0), it means that all phenotypic variation in the population is attributable to non-additive genetic influences and environmental factors. As a result, (VA ≈ 0) does not mean that the trait has no genetic basis; rather, it implies that the observed trait variation within the population under consideration cannot be explained by additive genetic variance [74]. Several previous studies of heritability for reproductive traits in some dairy cattle breeds, including Thai–Holstein crossbred cattle, are presented in Table 1.

The heritability values for reproductive traits in both Thai–Holstein crossbreeds and the other dairy cattle were low (Table 1). The difference in heritability values is due to different genetic and phenotypic variations in each population. The heritability of AFC, NSPC, CFI, IFL, FSCR, CR, DO, P90, and CI ranged from 0.01 to 0.08. The low heritability values mean the genetic component of these traits was low, and most of the variation of reproduction traits in these populations is due to management. In addition, a low value of heritability indicates that genetic improvement for these traits would be slow.

Faster genetic gain might be achieved by enhancing the accuracy of the heritability estimate, especially for reproductive traits. Berry et al. [81] reported that the on-farm usage of low-cost genomic technologies could help in a more precise parentage assignment, enhancing not only heritability but also genetic assessments. Poor record keeping with incorrect phenotypic data and pedigree could be a serious factor that can reduce the accuracy of the analysis. As a result, the recording system must be improved. More detailed records would allow researchers to partition the genetic effects between traits and result in more accurate fertility evaluations. In addition, standardized environmental circumstances can increase heritability by decreasing non-genetic variations among animals. Modern milking facilities, improved nutrition, and well-trained management personnel increased the possibility of genetic improvement in reproductive traits [74].

According to Table 1, the most popular model used in analyzing the variance components for these reproductive traits was MTM. Guo et al. [82] and Karaman et al. [83] reported that MTM outperformed STM for traits with poor heritability and a small number of records because phenotypic information for all traits of interest is not always accessible for all animals in a reference population; this is critical in practical breeding efforts. For example, there is typically a limited quantity of data for difficult or expensive traits to assess, such as carcass quality, feed efficiency, and disease traits. For traits with limited phenotypic data, the accuracy of EBV or GEBV derived via an STM will be low. An MTM will increase the accuracy derived by including information from associated and more readily measurable traits. Prediction using the MTM model gives more reliable results. Budhlakoti et al. [84] also reported that MTM has great potential to increase genetic gain and contributes to estimating the breeding value more precisely.

### 5.2. Genetic Correlation (rg)

Genetic correlations measure how closely two traits are related genetically [85]. Pleiotropy of genes is the major cause of the association, although linkage disequilibrium can also play a role. Pleiotropy refers to a gene’s ability to alter more than one character [74]. Genetic correlations are determined from the additive genetic variance and covariance between traits, $r_{g}=cov_{g}\;\left(X,\;Y\right)\;/\;\sqrt{V_{gX}V_{gY}}$, where X,  Y are traits, and $V_{gX}$, $V_{gY}$ are the variances of the traits.

Genetic correlations ranged from −1.0 to 1.0, and they can be positive or negative. The presence of positive or negative coefficients merely denotes whether the relationship is direct or inverse. Genetic correlations reveal how traits “covary” or change in tandem. When genetic correlations are near zero, each attribute is controlled by a distinct set of genes, and selection for one trait has minimal impact on the other. If the genetic correlation is positive, selection for one characteristic will increase the other, while selection for the other trait will reduce it if the genetic correlation is negative [74]. Genetic correlations are used by farmers to increase accuracy, minimize assessment time, and broaden the area of genetic evaluation [75]. The genetic correlation between reproduction traits in Thai–Holstein crossbred cattle is presented in Table 2.

Genetic correlations values in this study ranged from 0.36 to 1.00 and are included in the high category (Table 2). Strong positive genetic correlations (≥90) were observed between NSPC-IFL, CFI-DO, CFI-CI, IFL-DO, IFL-P90, and DO-CI. High positive genetic correlation values (0.36–0.89) were found between NSPC-CFI, NSPC-DO, NSPC-CI, CFI-IFL, IFL-CI, and FSCR-P90. Similar findings for NSPC-IFL, IFL-DO, and DO-CI were reported by González-Recio and Alenda [58] in Spanish dairy cattle: 0.91, 0.99, and 0.99, respectively. The values of genetic correlation between DO and CI in this study were also similar to those reported by Rahbar et al. [14] and Ghiasi et al. [86], of 0.98 and 0.99, respectively. The values between NSPC-CFI and CFI-IFL in Thai–Holstein crossbred cattle were higher than Spanish dairy cattle reported by González-Recio and Alenda [58] of 0.11 and 0.50, respectively. Differences in the additive variance and covariance of each trait in the population might explain the disparity in these results. High positive values imply that these reproductive traits are nearly genetically identical and are affected by the same genes. This means that the selection between traits with positive genetic correlations would be followed by an increase in other traits.

The high negative values of genetic correlations (>−0.53) in Thai–Holstein crossbred cattle were found between NSPC-FSCR, NSPC-P90, CFI-FSCR, CFI-P90, IFL-FSCR, FSCR-DO, FSCR-CI, DO-P90, and P90-CI (Table 2). A negative value of genetic correlations indicates that these traits have an inverse relationship with one another. For example, selection for a lower NSPC, for example, might result in a higher number of FSCR and P90, and a selection within traits that have a negative genetic correlation value will also be the same.

A study of Thai–Holstein crossbred dairy cattle also found a positive genetic correlation between reproductive traits (P90) and milk yield (MY) of 0.69 [50]. This result revealed that the selection of the P90 trait would increase milk production. The other study on tropical dairy cattle in Ethiopia showed negative genetic correlations between AFC-MY, CI-MY, and DO-MY of −0.24 ± 0.11, −0.10 ± 0.13, and −0.02 ± 0.14, respectively [87]. Conversely, the negative genetic correlation between CI-MY and DO-MY suggests that when milk output increases, fertility can be improved in this herd.

### 5.3. THI and Genetic Model for Heat Stress

#### 5.3.1. THI Model

THI is calculated from temperature (°C) and relative humidity. Several formulas in the previous studies reported by Nascimento et al. [88] were compared to calculate the THI; however, this was the best model that produced the most accurate results according to Berman et al. [5]:THIs = 3.43 + 1.058 × T − 0.293 × RH + 0.0164 × T × RH + 35.7 
where T is the average daily temperature (°C) and RH denotes the average daily relative humidity (%). Then, as a dummy variable, f(THI), a function of THI, was developed to quantify the reduction in reproductive performance under heat stress situations.
$$f\left(\mathrm{THI}\right)=\left\{\begin{array}{c} 0\;if\;THI\;\leq \;THI_{threshotd}\\ THI-THI_{threshold\;}\;if\;THI>THI_{threshold} \end{array}\right.$$

$THI_{threshold}$ were investigated at several different critical levels or threshold points, and the optimal model was determined by the lowest 2logL Akaike information criterion (AIC) and Bayesian information criterion (BIC) [15,89,90].

#### 5.3.2. Genetic Model

Several models were applied to evaluate variance components for heat tolerance in dairy cattle (Table 1). Sigdel et al. [15] and Aguilar et al. [89] purposed a multi-trait linear-repeatability test-day model (REP) to analyze the variance components for conception per insemination (binary trait) and multiple lactations at heat stress levels. Furthermore, Aguilar et al. [89] utilized the multiple-trait model (MTM) on DIM and f(THI) with the same effects as the REP model. Boonkum et al. [9] also used the MTM to estimate the variance components of DO. In agreement with Boonkum et al. [9], Oseni et al. [91] purposed the MTM and random regression model (RRM) to analyze the DO. In another study, Rahbar et al. [14] used an animal linear mixed model and univariate threshold models to analyze variance components for FSCR, gestation length (GL), NSPC, insemination outcome (IO), CI, calving birth weight (CBW), and DO. The (co)variance structure of each study was created according to the model, objectives, and problems of each study.

Some software is used by researchers to support the data analysis of each model. Aguilar et al. [89] used GIBBS2F90 to conduct their MTM on multiple lactations; the software implements Gibb’s sampling with a joint sampling of random correlated effects and traits [92]. Boonkum et al. [9] and Oseni et al. [91] used the SAS’s GLM method and AIREMLF90 program for their analysis of DO. Rahbar et al. [14] also used AI-REML in the DMU software package [93] to estimate variance components of reproductive traits using their genetic model.

## 6. The Approach of Genetic Selection for Reproductive Traits in Dairy Cattle

### 6.1. Traditional Breeding Method

The goal of livestock breeding is to select animals that will produce superior animals with higher yields [94]. The traditional breeding method (TAB) used the information of phenotypic records or pedigree for animal selection. This method is complicated for traits with low heritability values, such as reproductive traits, because they are more difficult to improve, and there are long generation intervals [95].

Furthermore, the new method was designed with the combination of the phenotypic and those of relatives into the estimated breeding value (EBV), called the best linear unbiased prediction (BLUP) [94]. This method successfully increases the genetic response to selection by improving the reliability of the EBV [96]. In addition, this method includes all systematic effects and exploits all pedigree information of the animals using a numerator relationship matrix to account for increases in additive genetic variance caused by inbreeding or assortative mating [97].

However, according to Ibtisham et al. [94], there are still a few limitations in using the BLUP method: (1) making accurate early selection decisions by regularly and timely recording of phenotypes is costly; (2) for some economics traits with low heritability, longevity traits, sex-limited traits, or expensive or difficult traits such as meat quality and disease resistance, it is less accurate and inefficient; (3) the breeding process is lengthy but necessary to collect sufficient phenotypic data to increase precision in performing the genetic evaluation.

Many models have been utilized for the genetic evaluation of reproductive traits, especially for interval traits, including the sire model (SM), sire–dam model (SDM), and the animal model (AM). However, a study comparison between these models revealed that AM could improve stability and accuracy in evaluating fertility traits in dairy cattle [98]. The SDM is comparable to the AM in terms of performance, although it is not significantly different. The AM has the benefit of delivering EBV of cows with intermediate reliability, and therefore it is better for interval traits.

The several traits of animal reproduction are observed as discrete outcomes. A threshold model has been proposed and applied to the problems of animal breeding which assumes that no observable normal variable is defined as a threshold. Evaluations based on threshold models for categorical traits showed that they could provide a greater response to selection than those derived from linear models [99].

### 6.2. Marker-Assisted Selection (MAS)

MAS is the combination between traditional breeding methods and molecular genetic methods. This method was introduced in 1900, with the concept of using specific markers on genes that affected economic traits in animals [94]. MAS requires accurate information about the relationship between genetic markers and related quantitative traits loci (QTL) for more precise results [100]. It provides an opportunity to predict the genotype of superior animals in the next generation, regardless of environmental conditions.

MAS has several advantages compared with the TAB method: (1) the process is faster since the phenotype of an individual may be forecasted at an early stage; (2) it is more profitable for sex-limited characteristics and difficult late-life traits; and (3) it is more useful for crossbreeding programmed by interrogating animals with superior genetics and better breeding value into local breeds [94]. However, the MAS technique uses a small number of loci that only account for a small amount of genetic variation, and can thus result in a small genetic gain [101]. Another limitation of MAS is that this method is quite expensive and economically inefficient when applied in commercial breeding programs [102]. The application of MAS in breeding programs is also limited because only a few markers have been proven to have a significant impact on animals’ economic traits, and they only account for a small percentage of genetic diversity [103].

Identifying and characterizing candidate genes and genetic variations associated with economically significant traits is essential in animal breeding. Each candidate gene has its own regulatory region or coding region [104]. Several potential genes associated with reproductive traits in dairy cattle were discovered in previous studies (Table 3).

### 6.3. Genomic Selection (GS) Method

Many SNPs covering whole genomes in animals have been discovered with genome-sequencing technology. Based on the genotypic information from whole genomes, a genomic selection (GS) method was proposed by using markers to estimate breeding values with high accuracy. Genomic selection is a useful tool and has been proven to raise genetic gain for low-heritability traits or difficult-to-measure traits [108]. Hence, this tool is suitable for reproductive traits in dairy cattle. Studies of the GS application strategies, the factors that influence GS accuracy, and the implementation of GS have been conducted in animal breeding [103].

In principle, GS predicts the breeding value of each genotyped individual using high-density (HD) markers that span the entire genome, which is subsequently referred to as genomic estimated breeding value (GEBV). Several approaches for determining the GEBV for cattle have been proposed, including a single-step Genomic Best Linear Unbiased Prediction (ssGBLUP) and a weighted ssGBLUP (WssGBLUP). A hybrid matrix (H) in the ssGBLUP is created by combining the pedigree-based relationship matrix (A) with the genomic relationship matrix (G), and when compared to multi-step genomic predictions, this advantage improves the accuracy and decreases the prediction bias of GEBVs [109]. However, the ssGBLUP does not assume the exact differences between the SNPs and their biological point of view; hence, the WssGBLUP method has been proposed [110]. The WssGBLUP accounts for locus-specific variance and uses various SNP weights when computing the G matrix [111]. These two approaches have been compared and contrasted in various studies and are commonly used in genomic prediction studies [112,113].

GS has the potential to overcome the limitations of MAS and more correctly forecast breeding values [114]. In MAS, only a few markers have been proven to significantly impact economically relevant traits. GS can fully utilize genotypic information from whole genomes in the genetic evaluation of animals, and breeding values can be predicted with high accuracy using genetic markers alone. GS is the latest technology and promises greater genetic advantages than TAB methods in dairy cattle. According to Gutierrez-Reinoso et al. [115], GS has higher reliability of 73.3% to 93.5% compared to TAB of 46% to 72%, depending on the number of progenies studied. GS also requires a relatively shorter time than a TAB to obtain progeny testing results. Although it requires more expensive equipment, the technology advances constantly, decreasing its cost. GS is more accurate and unbiased for modeling with unknown parent groups to predict the genetic merit [115]. For genetic variance estimation using pedigree matrices, GS produces high-reliability values for young bulls without pedigree or known sires [115]. GS also produces a high genetic gain of 50–100% for yield traits [116].

The application of GS in dairy cattle significantly increased the economic efficiency of animal breeding programs [117]. The benefits of GS in the dairy industry are generally due to a shorter generation interval, a reduction in costs for progeny testing, and an increase in the accuracy of EBV and genetic gains [118,119]. According to García-Ruiz et al. [116], the introduction of GS decreases the generation interval, especially for the sire(s) of bulls and dam(s) of bulls path from 7 to 2.5 years and from 4 to 2.5 years, respectively. In addition, Lund et al. [120] and Wiggans et al. [121] stated that the application of GS might improve genomic prediction reliability by 0.8 for production traits and 0.7 for fertility traits. Positive genetic trends also occur in low-heritability traits such as the productive life, daughter pregnancy rate, and somatic cell score [116]. Therefore, the benefit of GS for reproductive traits or traits with low heritability is high compared to TAB and MAS.

#### 6.3.1. Bayesian Approaches

The data are modeled at two levels using Bayesian estimation. The first is a model at the data level, while the second is a model at the variances of the chromosomal segments [122]. Several Bayesian statistical approaches have been applied in genomic assessment, with the hypothesis of marker effect distributions differing. Meuwissen et al. [108] presented distinct a priori distributions between the Bayes A and Bayes B approaches for simulating the variances of the effects of the markers.

The Bayes A technique assumes that the variance-of-marker effect varies across loci. The scaled inverted chi-square distribution is used to simulate the variances, χ^−2^ (v, *S*), where *S* denotes the scale parameter, and v denotes the degree of freedom. This is a valuable option since the resultant posterior distribution is similarly a scaled inverted chi-square when the information from the previous distribution is joined with the information from the data [108].

The Bayes B approach assumes variances of SNP effects in a genomic assessment context, with many SNP contributing to zero effects and a few contributing to substantial impacts on the trait [122]. Meuwissen et al. [108] presented a model in which a fraction of the markers have no impact (arbitrarily set at 0.95). Because some markers have a high likelihood of having zero variance, Gibbs sampling cannot be utilized to estimate the effects and variances of the Bayes B model. According to Meuwissen et al. [108], the Bayes B approach is commonly regarded as the “benchmark” in terms of genome prediction, although it is exceedingly time consuming in terms of computing.

Habier et al. [123] expanded the Bayesian alphabet by introducing the Bayes Cπ approach, in which the effect of SNP markers on π is considered to be zero, and a normal prior distribution is assumed for the variances relevant to the remaining (1 − π) markers, and the mixing parameter, π, may be inferred from the data rather than presumed to be known. Other approaches, such as the Bayesian LASSO reported by de los Campos et al. [124], assume that a double exponential prior distribution has also been utilized in animal breeding. Furthermore, Erbe [125] presented the Bayes R approach, which assumes a combination of normal prior distributions and accommodates SNPs with zero, small, medium, and high effects more effectively. In instances where one or more QTL with moderate or high effects exist, Bayesian regression approaches outperform GBLUP by a small margin, but when the trait is impacted by many genes, each with a small effect, GBLUP performs as well as or better than the Bayesian regression model [126].

Bayesian methods have some drawbacks. According to Legara et al. [127], the SNP effects might be modeled using a Bayesian approaches if the individual SNP variances were known and kept in the diagonal matrix B. This is different from the single-step genomic model, which allows for flexible modeling of SNP effects in terms of the number and (co)variance structure of fitted SNP markers. Moreover, for Bayesian SNP models assuming heterogeneous SNP variances for the various qualities, it is no longer easy to deduce correlations of traits at the SNP level from those at the genome level. This makes the Bayesian models more challenging to extend to multiple traits than the BLUP SNP models [127].

#### 6.3.2. Multi-Step Genomic Evaluation

Several strategies have been developed since genomic selection was initially proposed in animal breeding. A multi-step procedure was introduced by VanRaden et al. [101] for GS in dairy cattle using field datasets. The steps in the multi-step procedure are: (1) estimating the traditional breeding values, (2) generating fictitious observations for genotyped animals (bulls) such as daughter yield deviations (DYD) and the degressed evaluation (DD), (3) determining the direct genomic values (DGV) for genotyped animals using approaches based on SNP effects or genomic connections with DYD or DD as phenotypic observations, and (4) constructing an index based on pedigree and genomic predictions to estimate GEBV by the formula:GEBV = w1PA + w2DGV − w3PI
where PA denotes the parent average, and PI denotes the parental index derived from genotyped animals’ pedigree relationships. Because only a small percentage of animals in a lineage are genotyped, the quantity of phenotypic data required to compute DGV is often less than that required to calculate PA [101]. In this scenario, using DGV to account for the normal sire–dam PA, as well as the PA computed from a selection of genotyped ancestors in an index, can assist to enhance genomic predictions [128]. Furthermore, a previous study found that expanding a genotyped group’s reference population had a greater influence on prediction accuracy than increasing the number of SNP markers. Pseudo-observation might be computed as degressed proofs (DRP) instead of DYD, which is time-consuming and involves considerable work [121].

The multi-step approach was well suited to situations where the datasets belong to separate organizations and are condensed into a limited number of animals to genotype [128]. This includes a population with large numbers of average bulls with many daughters. The multi-step method has some limitations. It is difficult to create a DRP that is free of double counting when the population comprises both males and females, especially when genotyping includes parents and their progeny [127]. Weighting factors, such as variance components or selection index coefficients, are downsides of this strategy, as is information loss [101,129,130]. Moreover, these disadvantages may negatively affect the use of marker-assisted selection, especially for cows [131]. According to Misztal et al. [128] the multi-step approach is rather complex, and it relies on the existence of animals (bulls) with high correct EBVs based on pedigree information in its first form. Conversely, this method also needs to construct pseudo-observations that become biased by genomic preselection when a population contains non-genotyped animals with phenotypes because of BLUP, and the move to a single step is necessary [132].

#### 6.3.3. Single-Step Genomic Evaluation

Single-step genomic BLUP (ssGBLUP) was designed to solve possible issues by combining phenotypes, by merging phenotypic, pedigree, and genomic links into a combined relationship matrix. Furthermore, by substituting the connection matrix with the combined matrix, pedigree-only studies may be converted to genomic analyses, and pseudo-phenotypes are no longer required. This method also allows for the simultaneous evaluation of non-genotyped animals and genotyped animals. It is a simple and more accurate method that can reduce the number of steps in the evaluation [109].

Compared to multi-step genomic predictions, the ssGBLUP combines the pedigree-based relationship matrix (A) and the genomic relationship matrix (G) into a hybrid matrix (H). This benefit boosts the accuracy and minimizes the prediction bias of GEBVs [109]. In addition, Patry and Ducrocq [132] reported that ssGBLUP accounts for preselection due to pedigree BLUP, which can minimize bias, and is different from multi-step that does not take this into account. However, the ssGBLUP does not assume the exact differences between the SNPs and their biological point of view; hence, the WssGBLUP method has been proposed [110]. The WssGBLUP accounts for locus-specific variance and uses various SNP weights when computing the G matrix [111]. These two methods have been evaluated in various studies and are commonly used in genomic prediction studies [112,113].

The previous study suggested that the ssGBLUP method is a better option than the WssGBLUP method for crossbreeding with polygenic traits [111]. When comparing WssGBLUP to ssGBLUP for less-polygenic variables, Lourenco et al. [133] found that WssGBLUP might be more accurate. WssGBLUP can be beneficial for phenotypes with a small number of causal genes. It assumes that the traits are affected by a finite number of markers [111]. Below are some interesting factors of ssGBLUP according to Leggara et al. [127]:All relatives of genotyped individuals are automatically accounted for, as are their performances.Fitting genetic data and estimating additional effects are performed at the same time (e.g., contemporary groups). As a result, there will be no data loss.Feedback: all of a genotyped individual’s relatives benefit from the greater accuracy.Extensions are simple. Because this is a linear BLUP-like estimator, it can be immediately applied to more complex models (multiple traits, threshold traits, test day records). Any model that can be fitted with relationship matrices may also be fitted with combined relationship matrices.

Previously, the single step method has been used to evaluate the genetics for production traits of Thai–Holstein crossbred cattle. In a recent study, Sungkhapreecha et al. [134,135] proposed the ssGBLUP and ssGREML method to investigate the effect of heat stress on genetics for milk yield traits. The results revealed that the ssGBLUP technique was more accurate than the BLUP method, while ssGREML results are not much different from the REML method in genomic predictions. However, Buaban et al. [136] generated GEBV using a single-step random regression test-day model (ssRR-TDM) in contrast to an estimated breeding value (EBV) based on the pedigree-based model by utilizing the RR-TDM approach. The results showed that ssRR-TDM raised individual accuracies by 0.22 and validation accuracies by 0.07 in comparison to RR-TDM, when only bull genotypes were employed. Individual accuracies also increased by 0.02 with cow genotypes, while validation accuracies increased by 0.06. Based on these findings, a single-step method is commonly used in Thai–Holstein crossbred cattle because it generates a greater accuracy for genomic prediction. It is also possible to use the technique in genetic evaluation for reproductive traits in the future.

#### 6.3.4. Accuracy of GS

GS could increase the accuracy of EBV testing at an early age [137]. The square root of the proportion of genetic variation captured by the SNP panel (q) and the precision with which the genetic effects captured by SNPs can be assessed are used to calculate the accuracy of GEBV (r) [138]. The size of the reference population, the heritability of the phenotypes in the reference dataset, and the number of different QTLs that impact the trait are additional factors [139].

GS is now being used on a broad scale in cattle; however, it is mostly focused on production traits [18,139]. The accuracy of GS for dairy traits in developed countries varies from 0.50 to 0.85 for production traits with medium to high heritability to roughly 0.20–0.50 for fertility and survival variables with lower heritability [140,141]. According to Hayes et al. [142], ssGBLUP has a greater accuracy of 0.69 and 0.60 for production traits (protein yield and protein percentage), respectively, than the multi-step approach of 0.67 and 0.54. In other studies related to milk yield and fat percentage, the accuracy of these steps is higher than multi-step methods [143,144]. Recently, in Thai dairy cattle, a GS study on milk production traits and somatic cell score using a single-step method revealed increased average individual accuracies by 0.22 [136].

Several studies on GS were reported in dairy cattle regarding fertility traits, including AFC, the non-return rate, and calving ease [143]. The accuracy of prediction for AFC using the ssGBLUP model was 0.299 in the Thai crossbred cattle [144] and 0.56 in the Gyr dairy cattle breed [145]. For the non-return rate, the accuracy of genomic prediction was 0.39 in Canadian Holstein cows [145]. A GS study on calving ease in Holsteins (Canadian) showed that the accuracy performance was 0.69 [109]. Most researchers use the ssGBLUP method for reproductive traits because it has higher accuracy. In a previous study, ssGBLUP had higher accuracy than multi-step analysis, 0.43 vs. 0.42, respectively [142].

#### 6.3.5. Challenges and Opportunities in Applying GS

Thailand has had a dairy cattle breeding development program for about 60 years. They have attempted to produce its tropical dairy breed based on Holstein Friesian (HF) and indigenous Thai cattle through selection and inter se mating. Improving reproductive performance and resistance to heat is one of the main objectives of this crossbreeding program. The central database system of the Bureau of Biotechnology for Livestock Production, Department of Livestock Development, hosts milk-production data, reproduction data, and pedigree information. Several studies on genetic evaluation of reproductive traits in Thai–Holstein crossbred cattle have been carried out using these data [9,50,146]. THI may be estimated using climate data acquired from the nearest meteorological center based on postal code to evaluate heat resistance [12,135]. The small number of prospective bulls selected from Thailand’s dairy population, however, might be a problem. This matter will limit the intensity of selection and inaccurate prediction of young animals if they are still using the traditional evaluation method. To increase the size of the reference population, Thai government’s bureau began collecting genotypes of daughter-proven bulls and phenotyped cows in 2015 [136]. These genotypic data could be used to estimate GEBV, which gives a higher accuracy value than traditional methods. Sungkhapreecha et al. [134,135] previously estimated GEBV using the ssGBLUP method for milk yield and heat tolerance in Thai–Holstein crossbreed cattle. The results indicated that the ssGBLUP technique’s predictions were more accurate than the BLUP method’s, with a ratio of accuracy of 32–54% [134]. There is still no research on GS for fertility traits in Thai dairy cows under heat stress; hence, this is an excellent opportunity, considering the availability of data from the Thai government’s bureau. The application of GS in Thai–Holstein crossbred cattle is expected to reduce generation intervals to simplify the selection process and reduce maintenance costs. Using the ssGBLUP method will also provide an overview of the GS research reference model for improving reproductive traits under heat stress, especially in tropical countries.

Most dairy farmers in Thailand (>80%) are smallholders with an average of 30 animals per farm [146]. Smallholder farmers have two key challenges: (i) lack of infrastructure and (ii) human capacity, notably in the areas of technological competency, data processing, and interpretation [147]. The small genetic trends in Thai dairy populations imply that farmer capacity to learn new technology and absorb new information for improving dairy performance and profitability may be limited [148,149]. To overcome these limitations, Thai dairy farmers will require a program that incorporates systematic training and ongoing assistance to increase milk production and earnings in a long-term way. Furthermore, strengthening collaboration is needed among academics, industry, farmers’ groups, the public sector, charitable organizations, and development agencies to advance the development and implementation of genetic improvement programs.

## 7. Conclusions

Genetic evaluations for reproductive traits have been carried out in dairy cattle, especially in Thailand. The genetic variance for reproductive traits in livestock is generally low, indicated by low heritability values (<0.1). However, there were moderate to strong genetic correlations between reproductive traits and between reproductive traits and production traits. THI should be included in the model for genetics evaluation under heat stress. Several approaches have been used for the genetic selection of reproductive traits, and GS may be the appropriate genetic technique for Thai dairy cattle. For reproductive traits in dairy cattle, the single-step genomic BLUP (ssGBLUP) technique is more accurate than the multi-step method. The implementation of GS for reproductive traits in Thai–Holstein cattle is interesting to study in the future because it has not been widely adopted.

## Figures and Tables

**Figure 1 vetsci-09-00163-f001:**
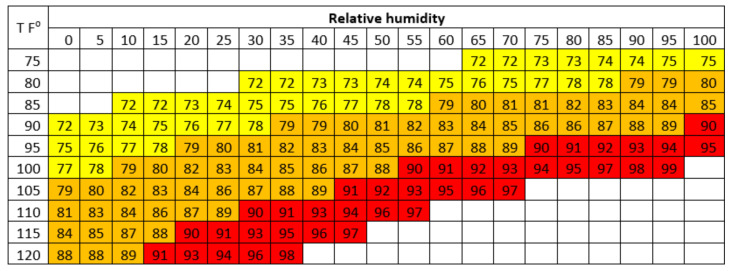
THI for dairy cows. THI 72 to 78 represents mild heat stress; THI 79 to 89 represents moderate heat stress; THI > 89 represents severe heat stress. (Source: Habeeb et al. [66]).

**Table 1 vetsci-09-00163-t001:** Heritability (h^2^) of reproductive traits in dairy cattle.

Traits ^1^	Breeds	h^2^	Model ^2^	References
AFC	Holstein (Brazil)	0.02	MTM	[76]
NSPC	Thai–Holstein crossbred	0.04	UM	[26]
	Spanish dairy cattle	0.02	STM	[58]
	Holstein (Iran)	0.04	ALM	[14]
	Swedish red	0.02–0.03	MTM	[77]
	Icelandic dairy cattle	0.02	MTM	[78]
CFI	Thai–Holstein crossbred	0.07	UM	[26]
	Spanish dairy cattle	0.05	STM	[58]
	Finland dairy cattle	0.07	UM	[79]
	Swedish red	0.02–0.03	MTM	[77]
	Nordic Holstein	0.05–0.06	REP	[80]
IFL	Thai–Holstein crossbred	0.02	UM	[26]
	Spanish dairy cattle	0.03	STM	[58]
	Finland dairy cattle	0.03	UM	[79]
	Icelandic dairy cattle	0.02	MTM	[78]
	Nordic Holstein	0.01–0.04	REP	[80]
FSCR	Thai–Holstein crossbred	0.01	UM	[26]
	Spanish dairy cattle	0.04	STM	[58]
	Holstein (Iran)	0.01	ALM	[14]
CR	Spanish dairy cattle	0.04	STM	[58]
	Icelandic dairy cattle	0.02	MTM	[78]
	Nordic Holstein	0.01–0.03	REP	[80]
DO	Thai–Holstein crossbred	0.07–0.08	MTM	[9]
	Spanish dairy cattle	0.04	STM	[58]
	Holstein (Iran)	0.02	ALM	[14]
	Holstein (Brazil)	0.03	MTM	[76]
P90	Thai–Holstein crossbred	0.02	TAM	[50]
	Spanish dairy cattle	0.06	STM	[58]
CI	Thai–Holstein crossbred	0.07	UM	[26]
	Spanish dairy cattle	0.04	STM	[58]
	Holstein (Iran)	0.03	ALM	[14]
	Holstein (Brazil)	0.03	MTM	[76]
	Icelandic dairy cattle	0.03	MTM	[78]

^1^ AFC, age at first calving; NSPC, number of services per conception; CFI, calving to first service interval; IFL, interval from first to last insemination; FSCR, first service conception rate; CR, conception rate; DO, days open; P90, pregnancy within 90 days after the first service; CI, calving interval. ^2^ Model: STM, single-trait model; MTM, multi-trait model; ALM, animal linear mixed model; UM, univariate model; REP, repeatability model; TAM, threshold animal model.

**Table 2 vetsci-09-00163-t002:** Genetic correlations within reproductive traits in Thai–Holstein crossbred cattle.

Traits ^1^	NSPC	CFI	IFL	FSCR	DO	P90	CI
NSPC		0.36	0.95	−0.80	0.64	−0.76	0.63
CFI			0.63	−0.58	0.90	−0.64	0.91
IFL				−0.93	0.90	−0.99	0.89
FSCR					−0.85	0.51	−0.85
DO						−0.53	1.00
P90							−0.96
CI							

^1^ NSPC, number of services per conception; CFI, calving to first service interval; IFL, interval from first to last insemination; FSCR, first service conception rate; DO, days open; P90, pregnancy within 90 days after the first service; CI, calving interval. Source: Buaban et al. [26].

**Table 3 vetsci-09-00163-t003:** Candidate genes that were previously associated with fertility traits in cattle.

No.	Traits ^1^	SNP ID	Chromosomes	Genes ^2^	Type of Mutation	References
1	FSCR	rs109137982	3	*FCER1G*	missense	[105]
		rs43745234	11	*FSHR*	missense	
		rs110789098	6	*IBSP*	missense	
		rs109629628	25	*PMM2*	missense	
		rs110660625	25	*TBC1D24*	missense	
		rs109956567	4	*LEP*	missense	[106]
		rs29004508	4	*LEP*	missense	
2	NSPC	rs137601357	7	*CAST*	missense	[105]
		rs109621328	7	*CD14*	missense	
		rs109137982	3	*FCER1G*	missense	
		rs41893756	18	*FUT1*	missense	
		rs109262355	20	*FYB*	missense	
3	DO	rs109669573	13	*BCAS1*	missense	
		rs110217852	6	*BSP3*	missense	
		rs137601357	7	*CAST*	missense	
		rs109621328	7	*CD14*	missense	
		rs109137982	3	*FCER1G*	missense	
		rs109956567	4	*LEP*	missense	[106]
		rs29004508	4	*LEP*	missense	
4	AFC	rs41256848	11	*LHR*	missense	[107]
		rs109956567	4	*LEP*	missense	[106]
		rs29004508	4	*LEP*	missense	
5	CFI	rs29004508	4	*LEP*	missense	[107]
6	CI	rs109956567	4	*LEP*	missense	[106]
		rs29004508	4	*LEP*	missense	
		rs29017188	8	*TLR4*	missense	

^1^ FSCR, first service conception rate; NSPC, the number of services per conception; DO, days open; AFC, age at first calving; CFI, calving to first service interval; CI, calving interval. ^2^
*FCER1G*, fc epsilon receptor Ig; *FSHR*, follicle-stimulating hormone receptor; *IBSP,* integrin-binding sialoprotein; *PMM2,* phosphomannomutase 2; *TBC1D24, TBC1* domain family member 24; *LEP,* leptin; *CAST,* calpastatin; *CD14, CD14* molecule *; FCER1G*, Fc epsilon receptor Ig; *FUT1,* fucosyltransferase 1; *FYB,* FYN binding protein; *BCAS1,* brain enriched myelin-associated protein 1; *BSP3,* binder of sperm protein 3; *PIT1,* pituitary transcript factor 1; *LHR*, luteinizing hormone/choriogonadotropin receptor; *TLR4*, Toll-like receptor 4.

## Data Availability

The data presented in this study are contained within the article.

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
