# Peer review of "An Appropriate Genetic Approach for Improving Reproductive Traits in Crossbred Thai–Holstein Cattle under Heat Stress Conditions"

_vetsci, 2022, doi:10.3390/vetsci9040163_

Round 1

Reviewer 1 Report

Heat stress is a huge concern to dairy cattle producers and stakeholders and the performance of crossbred Thai cattle is of interest to the international community. Unfortunately the paper lacks focus and often presents facts with little interpretation or context, which undermines the value of a review paper. Similarly, the value in this paper would be in reviewing the performance of Thai crossbreds and providing recommendations for their genetic improvement specifically (with consideration of heat stress). Many reports already exist on optimizing reproductive genetics in cattle (just from a quick search - Berry et al https://doi.org/10.1017/S1751731114000743, Shao et al https://doi.org/10.3389/fgene.2021.617128, Fleming et al https://doi.org/10.5194/aab-61-43-2018 ), robust reviews comparing genetic improvement strategies have also been published (Weigel et al https://doi.org/10.3168/jds.2017-12954) and even strategies for genetic improvement as they relate to complex, low heritability traits (Miles and Huson https://doi.org/10.3168/jds.2020-18297). Reviews also already exist on optimizing reproductive performance with heat stress (Negron-Perez et al https://doi.org/10.3168/jds.2019-16718), and even in breeding for heat stress resilience (Carabano et al https://doi.org/10.2527/jas.2016.1114). I'm afraid the paper in its current form does not contribute to the body of literature in a meaningful way, but could be restructured to provide insight and specific recommendations for improving the Thai crossbred population. To my knowledge no paper addressing that challenge exists and would be very valuable to the scientific community. Specific challenges with Thai cattle performance should be identified and the way these already widely-reviewed approaches can contribute to solutions should be discussed. 

General comments: nearly every sentence in this paper contained English language errors which made its interpretation very difficult. Many statements regarding genetics (perhaps due to the English) were incorrect (see detailed comments below) or too vague to be understood. Free software like Grammarly (https://www.grammarly.com/) can help with this and most universities have English editing/proofreading programs. Figure 1 was entirely missing from the paper. 

Please see specific comments pasted below:

Title is incomplete? Rephrase “Appropriate Genetic Approach for Improving Reproductive Traits in Thai Holstein Crossbred Cattle under Heat Stress Conditions”

L13 “causes heat stress” – please edit throughout paper for past/present tense

Global climate change is not the main cause of heat stress, but it does exacerbate the issue

L48-50 I’m not sure what the point here is. Genetic improvement cannot be the only long-term strategy, identifying and breeding for animals that possess natural resilience to high heat and humidity conditions must be done in concert with evolving management strategies.

L60 how was this temperature index calculated? Is it the same as THI, or including temperature only? Please standardize your terminology throughout the paper

L73 – what do you mean by the availability of SNPs? Alternative method to what? Traditional animal breeding strategies do not consider any genomic information and cannot link traits to regions of the genome.

L76 – the appropriate genetic approach for “improving”

L151 – what are the estimated losses on Thai farms? This is the novel content that would interest the reader 

L178 – please elaborate on CR. Most Interbull-participating countries calculate these separately as Sire CR, Cow CR, and Heifer CR. See https://interbull.org/ib/geforms

L230 – please elaborate on what the actual differences in skin morphology are

L237 – Figure 1 is missing from the paper?

L265-270 – this is a poor explanation of heritability and additive genetic variance. Heritability is a measure of the proportion of variance in the phenotype attributable to variance in the genotype – but heritabilities for a single trait can range widely depending on the traits and allele frequencies observed in your study population. What about broad-sense versus narrow-sense heritability? Why discuss additive models but not mention dominant and recessive modes of inheritance? What is your study population? Did all of these papers use the same breeds from the same countries?

L281 – this is not true. The heritability is low because the genetic component of these traits IS low. Other explanations for it being exceptionally low would be poor standardization and reporting of phenotypic records which is probably the biggest challenge for genetic improvement and this should be discussed.

L285-86 – why would “unexplained genetic effects” result in low heritability. Heritability does not offer any explanation… it is a simple ratio of genetic variance to phenotypic variance so your statement is incorrect.

L288 – what does FPR have to do with fertility?

L290 – you should explain why multi-trait models often outperform single trait models, and how they can be advantageous given the high intercorrelation of reproductive phenotypes

L300 – genetic correlation is essentially describing pleiotropy and explaining the proportion of heritability that can be shared by two traits, This definition needs to be improved.

L302 – all genetic correlations are linear, positive or negative coefficients are simply indicating whether it is a direct or inverse relationship.   

Table 2. It is to be expected that traits that are inverses would have high negative genetic correlation… Genetic correlations pulled from so many different papers should not be reported in a correlation matrix like this. Genetic correlations are going to change based on your study population and if these papers all had different study populations there is no internal or external validity to presenting these values together like this. What breeds are these from, what is the population you are trying to describe?

Table 3. Same comment about, though this presentation is more valid if I understand from your reference column that each row corresponds to a different study.

L360 – this discussion needs to be expanded upon. Strong negative genetic correlations between production traits and reproductive performance are a huge challenge for genetic improvement. The issue isn’t that “the same genes influence some reproductive and performance traits” is the nature of that influence, especially when they are at odds with each other

L407-409 – low heritability traits are difficult to be improved, period, regardless of TAB or GS. The advantage to GS is in reduced generation interval for more rapid progress and better resolution regarding the specific genetics affecting the trait you are interested in.

L456 – what does this sentence mean? All genes have coding regions (exons). Candidate genes do not always have functional annotation obviously related to the trait https://www.sciencedirect.com/science/article/pii/S0092867417306293

L489 – these has not be adequately explained. How exactly are MAS and GS different, and what problems with MAS can GS overcome?

Author Response

Response to Reviewer 1 Comments

Reviewer’s comments and suggestion:

Heat stress is a huge concern to dairy cattle producers and stakeholders and the performance of crossbred Thai cattle is of interest to the international community. Unfortunately the paper lacks focus and often presents facts with little interpretation or context, which undermines the value of a review paper. Similarly, the value in this paper would be in reviewing the performance of Thai crossbreds and providing recommendations for their genetic improvement specifically (with consideration of heat stress). Many reports already exist on optimizing reproductive genetics in cattle (just from a quick search - Berry et al https://doi.org/10.1017/S1751731114000743, Shao et al https://doi.org/10.3389/fgene.2021.617128, Fleming et al https://doi.org/10.5194/aab-61-43-2018 ), robust reviews comparing genetic improvement strategies have also been published (Weigel et al https://doi.org/10.3168/jds.2017-12954) and even strategies for genetic improvement as they relate to complex, low heritability traits (Miles and Huson https://doi.org/10.3168/jds.2020-18297). Reviews also already exist on optimizing reproductive performance with heat stress (Negron-Perez et al https://doi.org/10.3168/jds.2019-16718), and even in breeding for heat stress resilience (Carabano et al https://doi.org/10.2527/jas.2016.1114). I'm afraid the paper in its current form does not contribute to the body of literature in a meaningful way, but could be restructured to provide insight and specific recommendations for improving the Thai crossbred population. To my knowledge no paper addressing that challenge exists and would be very valuable to the scientific community. Specific challenges with Thai cattle performance should be identified and the way these already widely-reviewed approaches can contribute to solutions should be discussed. 

General comments: nearly every sentence in this paper contained English language errors which made its interpretation very difficult. Many statements regarding genetics (perhaps due to the English) were incorrect (see detailed comments below) or too vague to be understood. Free software like Grammarly (https://www.grammarly.com/) can help with this and most universities have English editing/proofreading programs. Figure 1 was entirely missing from the paper. 

Response to reviewer 1

Study about Thai Holstein crossbred cattle is interesting, especially for improving genetic reproductive performance under heat stress. here has been no publication regarding this, so it is interesting to discuss. According to your suggestion, we try to focus on discussing the topic of Thai dairy cattle, particularly in the section on genetic parameters. We highlighted revisions in green color in manuscript

Feedback for comments:

Point 1: Title is incomplete? Rephrase “Appropriate Genetic Approach for Improving Reproductive Traits in Thai Holstein Crossbred Cattle under Heat Stress Conditions”

Response 1: I have added the word “Improving” in the tittle [ line 2 -3]

Point 2: L13 “causes heat stress” – please edit throughout paper for past/present tense

Global climate change is not the main cause of heat stress, but it does exacerbate the issue

Response 2: I have changed the sentence into “As global temperatures have risen, heat stress in dairy cattle has become a severe concern for the worldwide dairy sector, especially in tropical countries” [line 29 – 30]

Point 3: L48-50 I’m not sure what the point here is. Genetic improvement cannot be the only long-term strategy, identifying and breeding for animals that possess natural resilience to high heat and humidity conditions must be done in concert with evolving management strategies.

Response 3: I have changed the sentence into “Improving genetics, identifying and breeding animals with a natural resilience to extreme heat and humidity, and establishing management approaches for long-term solutions should be done in concert”. [line 51 - 53]

Point 4: L60 how was this temperature index calculated? Is it the same as THI, or including temperature only? Please standardize your terminology throughout the paper

Response 4: I have changed the sentence into “A THI value of 72 for production and roughly 68 for reproduction was considered to be the threshold for heat stress” [line 62 - 63]

Point 5: L73 – what do you mean by the availability of SNPs? Alternative method to what? Traditional animal breeding strategies do not consider any genomic information and cannot link traits to regions of the genome.

Response 5: I decided to delete the sentences because it not corelated with the topic

Point 6: L76 – the appropriate genetic approach for “improving”

Response 6: I added the word “improving” to theses sentence [line 76]

Point 7: L151 – what are the estimated losses on Thai farms? This is the novel content that would interest the reader 

Response 7: I added the information about the cost for maintenance reproduction performance in Thai dairy cattle

“Dairy farmers of 44 farm in Central Thailand have to spend as much as 268,088 baht (8.220.37 USD) for improving the reproductive performance program every production period” [line161-163]

Point 8: L178 – please elaborate on CR. Most Interbull-participating countries calculate these separately as Sire CR, Cow CR, and Heifer CR. See https://interbull.org/ib/geforms

Response 8: I added the information about CR

“Most Interbull-participating countries divide CR into three categories: heifer conception rate (HCR), cow conception rate (CCR), and daughter pregnancy rate (DPR). HCR is defined as the percentage of inseminated heifers that become pregnant at each service [line 189 – 913]

Point 9: L230 – please elaborate on what the actual differences in skin morphology are

Response 9: I have added the information about the skin morphology

“Previous studies have found effects between different breeds of Sahiwal (B. indicus), HF (B. taurus), crossbred of HF75% and HF87.5 %, and the interaction effect between skin color and the genetic fraction of HF on skin morphology [65]. The study revealed that Sahiwal had the highest density and volume of sweat glands, as well as the high-est density of hair follicles (1,058 glands/cm2 ; 1.60 μ3 × 10−6) compared with pure HF (920 glands/cm2; 0.51 μ3 × 10−6 and crossbred of HF75% and HF87.5 % (709 glands/cm2; 0.68 μ3 × 10−6; and 691 glands/cm2; 0.61 μ3 ×10−6) respectively (P<0.01). However, the components of skin blood flow, such as capillary diameter, capillary circumference, and capillary surface, were higher in the HF purebred (8.33 μm; 26.48 μm; 2.07 μm per cm2, respectively), HF87.5% (7.13 μm, 22.40 μm, 1.95 μm per cm2, respectively), and HF75 % (7.85 μm; 24.92 μm; 1.83 μm per cm2, respectively) than in the Sahiwal (7.24 μm; 22.49 μm; 1.79 μm per cm2) (P<0.01). This suggests that skin morphology may in-fluence cutaneous evaporative heat loss capabilities and heat tolerance in crossbred cattle [line 234 – 247]

Point 10: L237 – Figure 1 is missing from the paper?

Response 10: No, it is not missing. It is in line 257

Point 11: L265-270 – this is a poor explanation of heritability and additive genetic variance. Heritability is a measure of the proportion of variance in the phenotype attributable to variance in the genotype – but heritabilities for a single trait can range widely depending on the traits and allele frequencies observed in your study population. What about broad-sense versus narrow-sense heritability? Why discuss additive models but not mention dominant and recessive modes of inheritance? What is your study population? Did all of these papers use the same breeds from the same countries?

Response 11: I have added the information about heritability

“Broad-sense heritability defined as the proportion of phenotypic variation (VP) due to genetic values (VG), that may include dominance (VD) and epistasis (VI) effects. Nar-row-sense heritability, on the other hand, only represents the proportion of genetic variation owing to additive genetic values (VA) [74]. The narrow sense heritability is commonly utilized in animal breeding because response to artificial (and natural) se-lection is dependent on additive genetic variation. Furthermore, additive genetic variation is the primary driver of relative similarity

Heritability ranges from 0 to 1. The higher value of heritability means that most of the differences among phenotypes of animals are genetic [75]. In contrast, while a low heritability indicates that VA is low, it says little about VG since genetic effects may be non-additive (VD and VI). As a result, a character h2 might still have a lot of genetic diversi-ty at the loci that contribute to the observed trait variance. When a variable has a heritability value of zero (0), it means that all phenotypic variation in the population is attributable to non-additive genetic influences and environmental factors. As a result, (VA = 0) does not mean that the traits have no genetic basis; rather, it implies that the observed trait variation within the population under consideration is completely due to environmental factors” [line 283 – 299]

Point 12: L281 – this is not true. The heritability is low because the genetic component of these traits IS low. Other explanations for it being exceptionally low would-be poor standardization and reporting of phenotypic records which is probably the biggest challenge for genetic improvement and this should be discussed.

Response 12: I have added the information about low heritability

“Heritability ranges from 0 to 1. The higher value of heritability means that most of the differences among phenotypes of animals are genetic [75]. In contrast, while a low heritability indicates that VA is low, it says little about VG since genetic effects may be non-additive (VD and VI). As a result, a character h2 might still have a lot of genetic diversity at the loci that contribute to the observed trait variance. When a variable has a heritability value of zero (0), it means that all phenotypic variation in the population is at-tributable to non-additive genetic influences and environmental factors. As a result, (VA = 0) does not mean that the traits has no genetic basis; rather, it implies that the observed trait variation within the population under consideration is completely due to environmental factors” [312 – 327]“

Point 13: L285-86 – why would “unexplained genetic effects” result in low heritability. Heritability does not offer any explanation… it is a simple ratio of genetic variance to phenotypic variance, so your statement is incorrect.

Response 13: I have deleted this statement

Point 14: L288 – what does FPR have to do with fertility?

Response 14: I have deleted the discussion about FPR in this paper because FPR is not included in reproductive traits although it plays a role in reproduction

Point 15: L290 – you should explain why multi-trait models often outperform single trait models, and how they can be advantageous given the high intercorrelation of reproductive phenotypes

Response 15: I have explained why MTM outperform ST

“Guo et al. [82] and Karaman et al. [83] reported that MTM outperformed STM for traits with poor heritability and a small number of records because phenotypic information for all traits of interest is not always accessible for all animals in a reference population; this is critical in practical breeding efforts. For example, there is typically a lim-ited quantity of data for difficult or expensive traits to assess, such as carcass, feed efficiency, and disease traits.  For traits with limited phenotypic data, the accuracy of EBV or GEBV derived via a STM will be low. A MTM will increase the accuracy de-rived by including information from associated and more readily measurable traits. [line 329 -327]

Point 16: L300 – genetic correlation is essentially describing pleiotropy and explaining the proportion of heritability that can be shared by two traits, This definition needs to be improved.

Response 16: I added the explanation about genetic correlations

“Genetic correlation represents how the genetic values for two traits are related. Genetic correlations are a measure of how closely two traits are related genetically [85].  Pleiotropy of genes is the major cause of the association, although linkage disequilibrium can also play a role. Pleiotropy refers to a gene’s ability to alter more than one character [74].

Genetic correlations ranged from –1.0 to 1.0, and they can be positive or negative. The presence of positive or negative coefficients merely denotes whether the relationship is direct or inverse. Genetic correlations reveal how traits "covary" or change in tandem. When genetic correlations are near zero, each attribute is controlled by a distinct set of genes, and selection for one trait has minimal impact on the other. If the genetic correlation is positive, selection for one characteristic will increase the other, while selection for the other trait will reduce it if the genetic correlation is negative” [line 342 – 354]

Point 17: L302 – all genetic correlations are linear, positive or negative coefficients are simply indicating whether it is a direct or inverse relationship.

Response 17: I added the explanation about genetic correlations

“Genetic correlations ranged from –1.0 to 1.0, and they can be positive or negative. The presence of positive or negative coefficients merely denotes whether the relationship is direct or inverse” [349-351]

Point 18: Table 2. It is to be expected that traits that are inverses would have high negative genetic correlation… Genetic correlations pulled from so many different papers should not be reported in a correlation matrix like this. Genetic correlations are going to change based on your study population and if these papers all had different study populations there is no internal or external validity to presenting these values together like this. What breeds are these from, what is the population you are trying to describe?

Response 18: I decided to change the table. I try to focus on discussing genetic correlations in Thai Holstein crossbred cattle. I replace table 2 with research data regarding the genetic correlation between reproductive traits in Thai Holstein crossbred cattle. [line 358 – 382]

Point 19: Table 3. Same comment about, though this presentation is more valid if I understand from your reference column that each row corresponds to a different study.

Response 19: I decided to delete this table because the data comes from research on various kinds of dairy cows. I focus on the study results in Thai Holstein crossbred cattle. [line 358 – 368]

Point 20: L360 – this discussion needs to be expanded upon. Strong negative genetic correlations between production traits and reproductive performance are a huge challenge for genetic improvement. The issue isn’t that “the same genes influence some reproductive and performance traits” is the nature of that influence, especially when they are at odds with each other

Response 20: I add some information about the genetics correlation between production and reproductive traits in Thai Holstein crossbred cattle

“A Study Thai Holstein crossbred dairy cattle also found a positif genetic correla-tion between reproductive traits (P90) and milk yield (MY) of 0.69 [50]. This result re-vealed that selection of the P90 trait will affect increasing milk production. The other study on tropical dairy cattle in Ethopia showed negative genetic correlations between AFC-MY, CI-MY and DO-MY of –0.24 ± 0.11, –0.10 ± 0.13 and –0.02 ± 0.14, respectively [87]. The negative genetic correlations between AFC and MY indicates that an increased selection based on MY would result in early onset of puberty. On the other hand, the negative genetic correlation between CI-MY and DO-MY suggests that when milk out-put increases, fertility can be improved to some extent in this herd” [line 385 – 393]

Point 21: L407-409 – low heritability traits are difficult to be improved, period, regardless of TAB or GS. The advantage to GS is in reduced generation interval for more rapid progress and better resolution regarding the specific genetics affecting the trait you are interested in.

Response 21: I change the sentence into “This method is complicated for traits with low heritability values, like reproductive traits, because they are more difficult to improve and due to long generation intervals” [line 438 – 440]

Point 22: L456 – what does this sentence mean? All genes have coding regions (exons). Candidate genes do not always have functional annotation obviously related to the trait https://www.sciencedirect.com/science/article/pii/S0092867417306293

Response 22: I change the sentence into “Identifying and characterization of candidate genes and genetic variations associated to economically significant traits is essential in animal breeding [487-488]

Point 23: L489 – these has not been adequately explained. How exactly are MAS and GS different, and what problems with MAS can GS overcome?

Response 23: I add some information about limitation of MAS

“GS has the potential to overcome the limitations of MAS and more correctly fore-cast breeding values [114]. In MAS, only a few markers have been proven to have significant impacts on economically relevant traits. GS could fully utilize genotypic information from whole genomes in the genetic evaluation of animals, and breeding values could be predicted with high accuracy using genetic markers alone.” [line 522 – 526]

Reviewer 2 Report

In my opinion, the presented manuscript  ‘Appropriate Genetic Approach for Reproductive Traits in Thai Holstein Crossbred Cattle under Heat Stress Conditions’  is an interesting review summing up the information about the current genetic approach of reproduction traits and heat stress conditions.  The review is a comprehensive and interesting study focused genetic analyzes in dairy cattle in the term of reproduction characteristics under heat stress. Manuscript is well written and included all necessary information. The text is well written, clear and easy to comprehend and follow. I believe that such review which sums up the genetic research on this topic is needed and expected. Thus, I recommend this manuscript to publication, after some minor revision.

Generally, the gene names should be in italics and it should be corrected throughout the whole manuscript.

Table 4 – Can you standardize the SNP nomenclature? I suggest showing the rs ID and on the other column adding the information about SNP type (missense, intron variant, splice variant), which will facilitate the interpretation of the influence of polymorphism on the trait.

Author Response

Response to Reviewer 2 Comments

Reviewer’s comments and suggestion:

In my opinion, the presented manuscript ‘Appropriate Genetic Approach for Reproductive Traits in Thai Holstein Crossbred Cattle under Heat Stress Conditions’ is an interesting review summing up the information about the current genetic approach of reproduction traits and heat stress conditions.  The review is a comprehensive and interesting study focused genetic analyzes in dairy cattle in the term of reproduction characteristics under heat stress. Manuscript is well written and included all necessary information. The text is well written, clear and easy to comprehend and follow. I believe that such review which sums up the genetic research on this topic is needed and expected. Thus, I recommend this manuscript to publication, after some minor revision.

Generally, the gene names should be in italics and it should be corrected throughout the whole manuscript.

Response to reviewer:

According to the reviewer suggestion, we changed the name of genes to be in italic in the manuscript, especially in the MAS section. We highlighted revisions in blue color in manuscript

Feedback for comments:

Point 1: Table 4 – Can you standardize the SNP nomenclature? I suggest showing the rs ID and on the other column adding the information about SNP type (missense, intron variant, splice variant), which will facilitate the interpretation of the influence of polymorphism on the trait.

Response 1: I have standardized the SNP name by using the rs ID. I have reduced some data because the SNP does not have and rs ID

  • I also add the information about SNP type [line 491]

Reviewer 3 Report

The review article (Appropriate Genetic Approach for Reproductive Traits in Thai Holstein Crossbred Cattle under Heat Stress Conditions) is purposed at reviewing and investigating different genetic evaluation approaches for reproduction traits in Thai Holstein crossbred cattle under heat stress conditions. The current review article is well-designed as heat stress conditions can affect reproduction traits and consequently genetic evaluation of these traits. Therefore publication of this manuscript can be reasonable for the Veterinary Science journal. Also, this article is well-written. however, there is a major concern which is described below:

I think authors must consider other genomic prediction methods (their advantages and disadvantages) in their review article. I wonder why the authors only considered ssGBLUP and WssGBLUP approaches?

 Some minor concerns and, numerous grammatical and typo errors are summarised below:

  1. Please remove “to” after “experienced” at line 30.
  2. Please change “especially for tropical countries” to “especially in tropical countries” in line 28.
  3. Remove “the” before “long-term” in line 49.
  4. In line 52 “response for heat stress” must change to “response to heat stress”.
  5. In line 67 remove “the” before “dairy”.
  6. In line 76 “the this”???
  7. In lines 84-85, change “be 818537 heads in 2021, with 358500 cows on 24764 farms” to “be 818,537 heads in 2021, with 358,500 cows on 24,764 farms”
  8. In line 109, “company' profitability” or company's profitability?
  9. Remove “as” in line 115.
  10. add a “,” after “In Thailand” in line 116.
  11. In lines 117-118, Please describe how AFC can affect farmer management related to determining age and body condition score (BCS). This might be informative for reviewers.
  12. In lines 126-127, How about their different genetic architectures?
  13. Please rewrite “Uterine infection has been reported could have effect on prolonged CFI and associated with poor reproductive performance in dairy cattle” in lines 1129-130.
  14. line 131, herdman???
  15. Lines 291-296, authors must discuss why MTM outperformed STM? Please provide reasons (for example by considering correlations of traits in MTM models). This might be informative for readers. Moreover, for MTM models in table 1, it is important how many traits were considered in their models? and what are they?
  16. Table 2 might make confuse readers. Breed/breeds? Their mothod?
  17. Line 294, change “give” to “gives”
  18. Line 256, change “time” to “times”.
  19. In line 331, “for for”?

Author Response

Response to Reviewer 3 Comments

Reviewer’s comments and suggestion:

The review article (Appropriate Genetic Approach for Reproductive Traits in Thai Holstein Crossbred Cattle under Heat Stress Conditions) is purposed at reviewing and investigating different genetic evaluation approaches for reproduction traits in Thai Holstein crossbred cattle under heat stress conditions. The current review article is well-designed as heat stress conditions can affect reproduction traits and consequently genetic evaluation of these traits. Therefore, publication of this manuscript can be reasonable for the Veterinary Science journal. Also, this article is well-written. however, there is a major concern which is described below:

I think authors must consider other genomic prediction methods (their advantages and disadvantages) in their review article. I wonder why the authors only considered ssGBLUP and WssGBLUP approaches?

Response to reviewer:

We considered ssGBLUP and WssGBLUP karena metode ini yang sekarang banyak digunakan oleh peneliti khususnya untuk GS. We highlighted revisions in yellow color in manuscript

Feedback for comments:

Point 1: Please remove “to” after “experienced” at line 30.

Response 1: I have changed remove the word “to [line 32]

Point 2: Please change “especially for tropical countries” to “especially in tropical countries” in line 28.

Response 2: I have changed the sentence into “As global temperatures have risen, heat stress in dairy cattle has become a severe concern for the worldwide dairy sector, especially in tropical countries” [line 29 -30]

Point 3: Remove “the” before “long-term” in line 49.

Response 3: I have removed “the” before “long-term” [line 51]

Point 4: In line 52 “response for heat stress” must change to “response to heat stress”

Response 4: I have changed it into “response to heat stress [line 52]

Point 5: In line 67 remove “the” before “dairy”.

Response 5: I have removed it [line 69]

Point 6: In line 76 “the this”???

Response 6: I have removed the word “this” [line 75]

Point 7: In lines 84-85, change “be 818537 heads in 2021, with 358500 cows on 24764 farms” to “be 818,537 heads in 2021, with 358,500 cows on 24,764 farms”

Response 7: I have changed it into “be 818,537 heads in 2021, with 358,500 cows on 24,764 farms” [line 83 – 84]

Point 8: In line 109, “company' profitability” or company's profitability?

Response 8: I have changed it into “company' profitability” [line 108]

Point 9: Remove “as” in line 115.

Response 9: I have removed it [116]

Point 10: add a “,” after “In Thailand” in line 116.

Response 10: I have added “a” after “in Thailand [line 117]

Point 11: In lines 117-118, Please describe how AFC can affect farmer management related to determining age and body condition score (BCS). This might be informative for reviewers.

Response 11: I have added discussion about how AFC can affect farmer management.

“However, AFC and body weight (BW) at calving have been found in a number of stud-ies to have a considerable impact on future milk production and herd survival [27–29]. A greater BW or body condition score (BCS) at calving improves milk production later [27]. In contrast, increased fat deposition excessively throughout the raising period, re-sulting in heavier heifers at first calving, may compromise mammary development, reducing future milk production [30]. The long-term impact of AFC on reproductive success is still unclear. Calving heifers between the ages of 25 and 26 months had shorter recurrent calving intervals than heifers between the ages of 24 and older age groups” [line 119 – 127]

Point 12: In lines 126-127, How about their different genetic architectures?

Response 12: I have added genetic as a factor affected CFI [line 135]

Point 13: Please rewrite “Uterine infection has been reported could have effect on prolonged CFI and associated with poor reproductive performance in dairy cattle” in lines 1129-130.

Response 13: I have added rewrited sentence “The other factor, a uterine infection, has been associated with extended CFI in dairy cattle and low reproductive performance” [ line 137]

Point 14: line 131, herdman???

Response 14: I have changed the sentence into “Poor management techniques such as detecting the presence of estrus in dairy cows can also contribute to long CFI” [line 138]

Point 15: Lines 291-296, authors must discuss why MTM outperformed STM? Please provide reasons (for example by considering correlations of traits in MTM models). This might be informative for readers. Moreover, for MTM models in table 1, it is important how many traits were considered in their models? and what are they?

Response 15: I have added the discussion about MTM Model

“Guo et al. [86] and Karaman et al. [87] reported that MTM outperformed STM for traits with poor heritability and a small number of records because phenotypic information for all traits of interest is not always accessible for all animals in a reference population; this is critical in practical breeding efforts. For example, there is typically a lim-ited quantity of data for difficult or expensive traits to assess, such as carcass, feed efficiency, and disease traits.  For traits with limited phenotypic data, the accuracy of EBV or GEBV derived via a STM will be low. A MTM will increase the accuracy de-rived by including information from associated and more readily measurable traits.” [line 329 – 337]

Point 16: Table 2 might make confuse readers. Breed/breeds? Their mothod?

Response 16: I decided to delete this table because the data comes from research on various kinds of dairy cows. I focus on the study results in Thai Holstein crossbred cattle. [line 358 – 368]

Point 17: Line 294, change “give” to “gives”

Response 17: I have changed it into “gives” [line 337]

Point 18: Line 256, change “time” to “times”.

Response 18: I have changed it into “times” [line 273]

Point 19: In line 331, “for for”?

Response 19: I have deleted the word “for”

Round 2

Reviewer 1 Report

This new version of the manuscript is much improved. The authors have made appropriate revisions to their paper in response to the reviewers, but they have still made some incorrect statements/inferences, explained below. The authors have refocused the first half of the paper to discussion of Thai crossbreds but that is lacking from the last half of the paper, where they give a general review of genomic selection methods. I do not see the value in section 6 without specifically refocusing it on Thai crossbreds. In its current form it is not a novel contribution to the literature, but re-summarizing topics which have already been reviewed extensively as I stated in my first report, with links to these reviews. The truly interesting information concerning Thai cattle (I only found 3 reports in section 6, L627-631, L656-658, 660-667) are lost in the body of the text. The scientific community does not need another review of these methods, but we do need a review of how they apply to Thai crossbreds. If, as the authors conclude “there is still limited research on GS for fertility traits in Thai dairy cows (L667)” and so there is little to discuss, instead they could perhaps give a recommendation for how this could be improved and talk about the challenges and opportunities in this research area.

The authors have added Figure 1 to the new manuscript at L267, it was missing from the original version I received.

L265 “reproductiveness” is not a word. I think you mean reproductive performance.

L308 this statement contradicts what you said directly preceding it. VA =0 would mean phenotypic variation cannot be explained by additive genetic variance, it does not mean it is completely due to environmental factors. Your statement L303 is not correct because VA does say a lot about VG, it just doesn’t not account for total genetic variance.  

L333-335 what does this sentence mean? I can’t understand what this scenario you are proposing is.. the problem with poor record keeping and reporting is that inaccurate or incomplete phenotypes are being used in genetic evaluation and more detailed records would allow researchers to partition the genetic effects between traits like pregnancy rate, conception rate, etc. and result in overall more accurate evaluations for fertility

L403-404 this is not true, negative genetic correlations are still linear.

L413-414 this is an unfair assumption and this inference should not be made. AFC is not directly related to puberty and we cannot connect those dots without more information about farm management and breeding criteria for those animals.

L600 males, not men

Author Response

Response to Reviewer 1 Comments

Reviewer’s comments and suggestion:

This new version of the manuscript is much improved. The authors have made appropriate revisions to their paper in response to the reviewers, but they have still made some incorrect statements/inferences, explained below. The authors have refocused the first half of the paper to discussion of Thai crossbreds but that is lacking from the last half of the paper, where they give a general review of genomic selection methods. I do not see the value in section 6 without specifically refocusing it on Thai crossbreds. In its current form it is not a novel contribution to the literature, but re-summarizing topics which have already been reviewed extensively as I stated in my first report, with links to these reviews. The truly interesting information concerning Thai cattle (I only found 3 reports in section 6, L627-631, L656-658, 660-667) are lost in the body of the text. The scientific community does not need another review of these methods, but we do need a review of how they apply to Thai crossbreds. If, as the authors conclude “there is still limited research on GS for fertility traits in Thai dairy cows (L667)” and so there is little to discuss, instead they could perhaps give a recommendation for how this could be improved and talk about the challenges and opportunities in this research area.

The authors have added Figure 1 to the new manuscript at L267, it was missing from the original version I received.

Response to reviewer 1

According to your suggestion, we add the new section (6.3.4) to discuss about the challenges and opportunities in applying GS especially for reproductive traits in Thai Holstein crossbred cattle (L700).

We highlighted revisions in green color in manuscript

Feedback for comments:

Point 1: L265 “reproductiveness” is not a word. I think you mean reproductive performance

Response 1: I have changed the word it to “reproductive performance” [line 256]

Point 2: L308 this statement contradicts what you said directly preceding it. VA =0 would mean phenotypic variation cannot be explained by additive genetic variance, it does not mean it is completely due to environmental factors. Your statement L303 is not correct because VA does say a lot about VG, it just doesn’t not account for total genetic variance.

Response 2: For L303 ,I have changed the sentence into “In contrast, while a low heritability indicates a low VA, it says a lot about VG, but it does not account for total genetic variance”[line 293-295].

For L 308, I have changed the sentence into “ As a result, (VA » 0) does not mean that the trait has no genetic basis; rather, it implies that the observed trait variation within the population under consideration cannot be explained by additive genetic variance” [line 298 -301].

Point 3: L333-335 what does this sentence mean? I can’t understand what this scenario you are proposing is.. the problem with poor record keeping and reporting is that inaccurate or incomplete phenotypes are being used in genetic evaluation and more detailed records would allow researchers to partition the genetic effects between traits like pregnancy rate, conception rate, etc. and result in overall more accurate evaluations for fertility

Response 3: I have changed the sentence into “Poor record-keeping with incorrecct phenotypic data and pedigree could be a serious factor that can reduce the accuracy of the analysis. As a result, the recording system must be improved. More detailed records would allow researchers to partition the genetic effects between traits and result in more accurate fertility evaluations”. [line 321-325].

Point 4: L403-404 this is not true, negative genetic correlations are still linear

Response 4: I have changed the sentence into “A negative value of genetic correlations indicates that these traits have an inverse relationship with one another” [line 381]

Point 5: L413-414 this is an unfair assumption and this inference should not be made. AFC is not directly related to puberty and we cannot connect those dots without more information about farm management and breeding criteria for those animals.

Response 5: I decided to delete this assumption because there is no information about farm management and breeding criteria for those animals

Point 6: L600 males, not men

Response 6: I have changed it to “males” [613]

Reviewer 2 Report

The authors have responded to all of the comments and suggestions and the manuscript has been improved significantly. I recommend presenting the manuscript for publication in the current version. 

Reviewer 3 Report

Authors applied most of my comments, however, there is steel my main concern as I mentioned in my previous report (“I wonder why the authors only investigated ssGBLUP and WssGBLUP approaches?”). However, ssGBLUP has been the most efficient method for genomic prediction in most of the former studies, it would be important that what are the benefits of ssGBLUP in comparison with multi-step methods such as Bayesian methods and GBLUP? Also, what are the benefits of applying ssGBLUP for genomic selection programs in crossbred Thai–Holstein cattle under heat stress conditions? Also, authors must answer my question in English, not Indonesian in “response to reviewer” section (here: We considered ssGBLUP and WssGBLUP karena metode ini yang sekarang banyak digunakan oleh peneliti khususnya untuk GS.)

Author Response

Response to Reviewer 3 Comments

Reviewer’s comments and suggestion:

Authors applied most of my comments, however, there is steel my main concern as I mentioned in my previous report (“I wonder why the authors only investigated ssGBLUP and WssGBLUP approaches?”). However, ssGBLUP has been the most efficient method for genomic prediction in most of the former studies, it would be important that what are the benefits of ssGBLUP in comparison with multi-step methods such as Bayesian methods and GBLUP? Also, what are the benefits of applying ssGBLUP for genomic selection programs in crossbred Thai–Holstein cattle under heat stress conditions? Also, authors must answer my question in English, not Indonesian in “response to reviewer” section (here: We considered ssGBLUP and WssGBLUP karena metode ini yang sekarang banyak digunakan oleh peneliti khususnya untuk GS.)

Response to reviewer:

We considered ssGBLUP and WssGBLUP because these methods are now widely used by researchers, especially for GS. Previous researchers have also used this method to evaluate the genetic for production traits in Thai–Holstein crossbred cattle. The result revealed that this technique generates a greater accuracy for genomic prediction; hence, I recommend this method for reproductive traits in the future [line 660-673].

According to your suggestion, I added more methodologies about Bayesian model and its limitation in section 6.3.1., hopefully, this will provide an overview of the comparison of the three methods. I also have mentioned the comparison between single-step and multi-step methods in this manuscript [line 631 -636] to provide insight into these two methods for genomic prediction. For the benefits of applying ssGBLUP in Thai–Holstein cattle I added the additional information in line 722-726. We highlighted revisions in yellow color in manuscript.

Round 3

Reviewer 1 Report

The manuscript is much improved and the new section on challenges and opportunities for GS applied to Thai cattle is a strong ending to this paper that provides valuable insights to the research community. 

Reviewer 3 Report

Now, I think the reviewed manuscript is publishable.